# Distribution of Larval Habitats and Efficiency of Various Trap Settings to Monitor Sympatric *Aedes albopictus* and *Aedes aegypti* in La Reunion

**DOI:** 10.3390/insects16090932

**Published:** 2025-09-04

**Authors:** Caroline Vitry, Ronan Brouazin, Anthony Herbin, Mathieu Whiteside, Cécile Brengues, Thierry Baldet, Renaud Lancelot, Jérémy Bouyer

**Affiliations:** 1Plateforme Technologique Cyroi, Umr Astre, Cirad, Inrae, University of Montpellier, 97490 Sainte-Clotilde, La Réunion, France; carolinevitry974@gmail.com (C.V.); ronan.brouazin@cirad.fr (R.B.); thierry.baldet@cirad.fr (T.B.); renaud.lancelot@cirad.fr (R.L.); 2Plateforme Technologique Cyroi, Umr Mivegec, Institut de Recherche pour le Développement (IRD), University of Montpellier, CNRS, 97490 Sainte-Clotilde, La Réunion, France; anthony.herbin@ird.fr (A.H.); mathieu.whiteside@ird.fr (M.W.); cecile.brengues@ird.fr (C.B.)

**Keywords:** mosquito monitoring, pyriproxyfen, boosted sterile insect technique, breeding site, chironomids

## Abstract

Two *Aedes* mosquitoes, vectors of dengue and chikungunya viruses, are found in Saint-Joseph (La Reunion): *Aedes albopictus* (the most widespread and abundant) and *Aedes aegypti*, with contrasted ecology. This study aimed to prepare for a boosted sterile insect technique control trial using sterile male *Aedes albopictus* treated with a larval biocide. We assessed the efficacy of two mosquito traps to catch these two species. Also, we investigated which breeding sites they shared. Using a standard comparison procedure, we showed that BG Sentinel adult traps baited with bottled carbon dioxide were equally attractive for both species. The survey of breeding sites showed that *Aedes aegypti* larvae can be found in vacoas (*Pandanus utilis*) located several hundred meters away from ravines. However, larvae of *Aedes albopictus* were found in most vacoas, thus reassuring that targeting this species in the control program would also allow for suppressing *Aedes aegypti* populations.

## 1. Introduction

The repeated occurrence of arbovirus outbreaks in La Reunion [1,2,3] highlights the need for effective tools to monitor and control *Aedes* mosquito populations. The two *Aedes* species considered to be the most important vectors of arboviruses worldwide are present in La Reunion: *Aedes albopictus* and *Aedes aegypti* [4]. *Aedes albopictus* is widely distributed in urban and rural areas, from coastal zones to elevations of up to 1200 m, and in both natural and artificial breeding sites. In contrast, *Ae. aegypti* populations are confined to a few natural environments, especially ravines in the south and west of the island. This limited distribution is attributed to historical indoor spraying campaigns against malaria vectors, which had a particular impact on the endophilic species *Ae. aegypti*, and ecological competition with *Ae. albopictus* being more competitive than *Ae. aegypti* in urban larval habitats ecological competition, as *Ae. albopictus* out-competes *Ae. aegypti* in urban areas [5,6]. Recurrent dengue outbreaks have been recorded in La Reunion in recent decades, with 29,577 confirmed cases and 33 deaths reported in 2021. The transmission of the dengue virus is now considered endemic [7,8]. In 2005–2006, during a large chikungunya epidemic, more than 300,000 cases were reported, with more than 200 deaths [2]. Furthermore, new cases of chikungunya were reported in August 2024, preceding a large outbreak of chikungunya during the austral summer of 2025 [9]. Although vaccines are now available against dengue and chikungunya viruses, mosquito control remains an important component of preventing and mitigating mosquito-borne infections.

The Sterile Insect Technique (SIT) is a promising and environmentally friendly approach [10]. It involves the mass-rearing, sterilization, and release of male mosquitoes that compete with wild males in mating with wild females. Because most female mosquitoes only mate once in their life [11,12], fewer offspring are produced when SIT is implemented. This method has been tested in field trials in La Reunion and Mauritius [13,14,15,16]. A derived method, boosted SIT, combines the release of sterile males with the application of a biocide (pyriproxyfen) to their body [17]. The treated males transfer the biocide to homospecific females during mating or other physical contacts. Subsequently, the contaminated females deposit the biocide in breeding sites, thereby preventing larval development and the emergence of adult mosquitoes. Cage trials have shown that breeding sites can be directly contaminated by sterile males coated with pyriproxyfen, even in the absence of females [18]. Because the two species share some of their breeding sites, they can be suppressed simultaneously. This heterospecific efficacy was observed during the pilot suppression trial conducted at the same site, where the release of sterile *Ae. aegypti* males treated with pyriproxyfen led to the suppression of more than 90% of the populations of this species, as well as approximately 60% of the populations of *Ae. albopictus* [19].

In all cases (SIT or boosted SIT), accurate monitoring of targeted mosquito populations is essential to assess suppression and provide useful indicators for the control program, such as the sterile to wild male ratio. Following small-scale field trials of SIT and boosted SIT, a phase III boosted SIT was started in 2025 in Saint-Joseph (South of La Reunion Island) within the OpTIS project, with releases of sterile *Ae. albopictus* males treated with pyriproxyfen. The OpTIS project aims to evaluate the efficacy of boosted SIT to suppress *Aedes* mosquitoes and to prevent dengue and chikungunya, as well as its acceptability for beneficiary populations and the absence of environmental impact. Before starting the control trial, we implemented this study to refine the parameters of the Biogents Sentinel (BGS) traps (Biogents, Regensburg, Germany) commonly used to monitor adult *Aedes* populations in La Reunion [20], and to evaluate the possibility of using ovi-sticky traps, widely used in Singapore to monitor both *Ae. aegypti* and *Ae. albopictus* [21,22,23]. We also carried out a survey to assess the presence of larvae of these two species and better predict the likely impact of releasing pyriproxyfen-treated sterile *Ae. albopictus* males on *Ae. aegypti* populations.

## 2. Materials and Methods

### 2.1. Traps

BGS traps consist of a foldable dark blue fabric container and a perforated white lid that covers its opening. Air is drawn into the trap by an electrical fan through a black tube and a black net acting as a cage. They were baited with carbon dioxide released from a bottle at 0.2 L/24 h as previously described [24]. We evaluated whether the attractiveness of this device was improved by adding BG Lure, a commercial attractant made of pellets that emit an odor that mimics human sweat https://sea.biogents.com/attractants/bg-lure-attractant/(accessed on 2 September 20025). Fresh pellets were left out for one week prior to their first use to avoid a repellent effect [24].

Ovi-sticky traps are small plastic black jars containing tap water and covered with an adhesive paper strip inside, thus catching any insect that enters the trap and lands or walks on the sticky paper. Tap water was dechlorinated: it was first poured into a dedicated container that was left open for 48 h at room temperature. We evaluated whether the attractiveness of this device improved when tap water was replaced with a solution prepared with 50 g of hay infused in 5 L of water for one week and then diluted with tap water in a ratio of 1 to 5.

### 2.2. Design

The four trap settings described above were compared using a Latin square design. Four traps with these settings were placed at four different positions and simultaneously permuted at each trapping, so that each trap occupied each position once at the end of a replicate. The positional and time variations were, therefore, the same for each trap. Two protocols were used, each with a different set of two locations (Figure 1):

Protocol 1: A ravine located in the eastern part of the study site, and an orchard located in its western part;Protocol 2: Two close sets located in an urban area located in the northern part of the study site.

Trap rotations were modified between the two protocols to increase the probability of detection and the abundance of catches.

Protocol 1: Every 24 h for BGS traps and ovi-sticky traps;Protocol 2: Every week, with BGS traps left in place for 48 h, and ovi-sticky traps for one week.

At each rotation, the adult insects captured were poured into a plastic container and stored at −20 °C to kill them. Then, they were poured into Petri dishes for identification using a binocular and a morphological key adapted to local mosquitoes. Individuals of *Ae. aegypti* and *Ae. albopictus* were counted by species and sex.

### 2.3. Larval Habitat

To assess the distribution of *Aedes* larvae across different habitat types, larval samples were collected from two different environments: (i) leaves stipes in the canopy of vacoa trees (*Pandanus utilis*, an endemic shrub) or (ii) domestic containers in private gardens such as flower pots, rainwater collection tanks, and pet bowls. The sampling sites were located at various distances from ravines or rivers where *Ae. aegypti* populations are considered restricted. The samples were collected during four field sessions: one in February (before the devastating Cyclone Garance), once in March, and two in April 2025. These time points were spaced at least 15 days apart to allow for the development of new cohorts and to assess potential temporal variations in the distribution of species related to environmental or meteorological conditions. The larvae were collected using 20 mL Pasteur pipettes. Five leaves per vacoa were sampled by inserting the pipette in the leaf stipe, where rainwater accumulates (15–50 mL was collected from each stipe where water was present, depending on previous rainfall). For domestic containers, larvae were collected from the surface of stagnant water. Once collected, the larvae were transferred to Petri dishes containing a small volume of water and frozen at −20 °C until identification. The *Aedes* L3–L4 instar larvae were identified with a binocular using an appropriate identification key for mosquito species [25].

### 2.4. Data Analysis

We performed separate analyses for *Ae. aegypti* and *Ae. albopictus* under the assumption of different drivers for their population dynamics, including different host patterns and contrasting breeding sites. We used a Bayesian mixed effect Poisson model of mosquito density to assess the effect of BG Lure as an additional attractant to CO_2_-baited traps, conditionally depending on the mosquito sex. This model included two fixed effects, coded as two main effects and their interaction; two dummy variables, coded as 0 (‘without lure’ or ‘female’) or 1 (‘with lure’ or ‘male’). In addition, an observation-level identifier was included as a random effect associated with the model intercept to account for the data overdispersion [26]. For observation *i* with count $y_{i}$ and expected mean $E\left(i\right)\;=\;\mu_{i}$, the linear predictor $\eta_{i}$ is as follows:$$\eta_{i}\;=\;log\left(\mu_{i}\right)\;=\;b_{0,i}\;+\;b_{1}\;\times \;sex_{i}\;+\;b_{2}\;\times \;lure_{i}\;+\;b_{3}\;\times \;sex_{i}\;\times \;lure_{i}$$
with $b_{0,i}\;=\;b_{0}\;+\;u_{i}$, and $u_{i}$ is the random effect with a Gamma prior (precision 0.0005); $b_{0}$, $b_{1}$, $b_{2}$, and $b_{3}$ are the coefficients of the fixed effect. The full model is as follows:$$y_{i}\;=\;e^{\eta_{i}}\;+\;\epsilon_{i}$$
with $y_{i}∼P\left(\mu_{i}\right)$, i.e., a Poisson distribution with mean $E\left(y_{i}\right)\;=\;\mu_{i}\;=\;e^{\eta_{i}}$, and $\epsilon_{i}$ is the error. In addition, the logarithm of the duration (in days) of each trapping session was used as an offset term in the mixed-effect Poisson model to account for differences in sampling effort between protocols 1 and 2. The model coefficients were fitted using an integrated nested Laplace approximation of the Markov chain process, implemented in the INLA add-in package for R [27,28]. Then 10,000 simulated fitted values were drawn in the posterior distribution of the model coefficients and stored in a matrix *M* with 96 rows (2 locations × 4 traps × 4 replicates + 2 locations × 4 traps × 8 replicates) and 10,000 columns (one column for each simulated dataset). Its rows were aggregated by covariate pattern (observed combinations of ‘sex’ and ‘lure’). Estimates of $\mu_{i}$ and its credible lower and upper 95% limits were obtained taking the mean and quantiles 2.5% and 97.5% of the aggregated rows. The matrix *M* was also used to estimate the relative densities $\theta_{i}$ and their 95% credible interval $\theta_{i}\;=\;\mu_{i}/\mu_{0}$, where $\mu_{0}$ is the density in the reference pattern (e.g., average count of females in traps without the lure). The 95% credible intervals for $\theta_{i}$ were used to assess the statistical significance of the additional attractiveness (that is, higher density) associated using lure with CO_2_-baited traps (α = 0.05): should this interval cover 1.00, the null hypothesis of no additional attractiveness is accepted; otherwise, it is rejected. In addition, the random effect $u_{i}$ was used to explore the variability of attractiveness according to the location of the trap, as well as the trend in time in the dynamics of the mosquito populations.

To assess the joint distribution of larvae in vacoa samples, their count was modeled with a negative binomial regression to account for the overdispersion in the data. With this maximum-likelihood method, overdispersion is modeled by a negative binomial distribution [29,30]. For a given larval site, the larval count *m* follows a Poisson distribution P(λ), where λ follows a Gamma distribution G with mean μ and shape parameter *k* and variance $Var\left[\lambda \right]\;=\;\mu^{2}/k$. Defining the overdispersion parameter Φ = 1/k, the marginal mean and variance of *m* are E[m] = μ and $Var\left[m\right]\;=\;\mu \;+\;\Phi \;\times \;\mu^{2}$.

In the generalized linear model framework [31], *m* can be modeled with covariates with fixed effect coefficients *b* and design matrix *X*: E(m) = μ = exp(X·b) = exp(η). We used two covariates, each with three categories: ‘species’ (reference = *Ae. albopictus*) and ‘month’ (month when the sample was made, reference = February). Firstly, we considered the two main effects and their interaction; then, we tried to find a better sub-model with a stepwise procedure based on the Akaike information criterion corrected for small samples (AICc) [32].

## 3. Results

### 3.1. Insect Catches

In total, 733 mosquitoes were caught during 96 trapping sessions from 3 February 2025 to 29 April 2025, including 447 *Aedes albopictus*, 187 *Aedes aegypti*, 66 unidentified *Aedes* spp. (related to one of the two previous species but too damaged to be identified), 26 *Culex quinquefasciatus*, and 7 chironomids (Figure 2). Among the catches, only 3 mosquitoes (female *Ae. albopictus*) were caught in ovi-sticky traps. We concluded that this type of ovi-sticky trap was not adapted for mosquito monitoring in La Reunion. For the remainder of the analysis, we focus on evaluating the effect of adding BG Lure to CO_2_-baited traps on their attractiveness.

### 3.2. Effect of Lure on CO_2_-Baited Traps’ Attractiveness

The results are shown in Figure 3.

Without lure, the attractiveness of the CO_2_-baited trap was the same for the females and the males of both species.With lure, the attractiveness of the CO_2_-baited trap was unchanged for the females of both species, as well as for male *Ae. aegypti*. In contrast, it was 4.3 times higher for male than for female *Ae. albopictus*: relative attractiveness RA = 4.3, 95% credible interval [2.2; 7.6].

### 3.3. Variations in Space and Time

The random effect *u*, associated with the model intercept, provided estimates—on the log scale—of presumably random fluctuations in relative attractiveness. For protocol 1 (Figure 4a), the estimated random effect $\hat{u}$ was lower in the ravine than in the orchard (*t* test, difference *d* = −1.0, *t* = −2.4, *p* = 0.02) and higher for *Ae. aegypti* than for *Ae. albopictus* (*t* test, *d* = 1, *t* = 2.4, *p* = 0.02). For both mosquito species, a sharp strong decrease in $\hat{u}$ was observed in six of eight traps between the start and the end of the survey (one week, early February 2025). More generally, there were large variations in $\hat{u}$ between and within the traps, for both species.

The random effect patterns for the data collected with protocol 2 (Figure 4b) were clearer than with protocol 1, in relation to the longer time period, as well as the different and less contrasted trap locations (Figure 1). On average, $\hat{u}$ was not significantly different for *Ae. albopictus* and *Ae. aegypti* (*t* test: *d* = −0.1, *t* = −0.3, *p* = 0.76). However, the variations between and within the trap in $\hat{u}$ were lower for *Ae. albopictus* than for *Ae. aegypti*. For the latter, sharp and positive peaks were observed on the same dates for three of eight traps (in two different locations). In addition, $\hat{u}$ appeared to decrease with time, with less variability for *Ae. albopictus* than for *Ae. aegypti*.

A linearly decreasing trend was observed in the distribution of the estimated random effect of the density Poisson models of *Ae. albopictus*, strongly correlated with the minimum daily temperature (Figure 5c): bilateral test, Spearman rank correlation, $\hat{\rho }\;=\;0.91$, s = 26, *p*$\;<\;2.210^{-16})$. The random effect pattern was very different for *Ae. aegypti*. It started close to 0, and then it consistently increased during one month, that is, shortly after the passage of Cyclone Garance. Then, it started to decrease in parallel with the daily minimum temperature.

### 3.4. Larvae

The larvae were collected during four sampling sessions between 14 February and 22 April 2025. A total of 101 larvae were collected from 10 breeding sites (four domestic containers and six vacoas): 72 *Ae. albopictus*, 6 *Ae. aegypti*, and 23 chironomids. *Aedes albopictus* was found in 9 of 10 sites (5 of 6 in vacoas) and *Ae. aegypti* in 5 of 10 sites (4 of 5 in vacoas); both were present together in 4 of 10 sites (3 of 6 in vacoas), and *Ae. aegypti* was found alone in 1 of 10 sites (1 of 6 vacoas). No *Culex* larvae were collected. All chironomids were found in vacoa breeding sites, as well as five of six *Ae. aegypti* larvae. Thus, *Ae. albopictus* was the highly predominant species in domestic containers. Furthermore, the number of larvae collected decreased with time: 54 in February (65% in vacoas, 33 samples), 37 in March (35% in vacoas, 33 samples), and 13 in April (85% in vacoas, 66 samples). Of the 10 sampling sites, *Ae. albopictus* was found at 5 sites (one of four domestic containers, four of six vacoas). *Aedes albopictus* was detected at four of five sites where *Ae. aegypti* was found. Chironomids were found in four of six vacoas. The number of larvae collected varied greatly between the samples. Up to 12 *Ae. albopictus* larvae were found in a unique sample, 2 *Ae. aegypti* larvae, and 5 chironomid larvae.

The results of the correlation analysis of the co-occurrence of larvae by insect species in the vacoas are shown in Figure 6. The model with the lowest AICc was the additive model. The count (density) of *Ae. albopictus* larvae in vacoa samples was negatively correlated with the same index for chironomids and *Ae. aegypti* in March and April. In contrast, the densities of chironomids and *Ae. aegypti* larvae were positively correlated, as well as the densities in March and April. The density of larvae in vacoa samples was lower for *Ae. aegypti* than for *Ae. albopictus* (*p* = 0.019), as well as in April compared to March (*p* = 0.001).

## 4. Discussion

The main objective of this study was to evaluate the efficiency of different trap types and baiting strategies for sampling populations of *Ae. albopictus* and *Ae. aegypti* mosquitoes in various environments in Saint-Joseph (La Reunion). We compared BGS traps baited with CO_2_ alone to those combining CO_2_ with BG-lure and ovi-sticky traps filled with hay infusion or water. In this trial, unbaited BGS traps were not considered because recent work in the same area showed that they were less attractive than CO_2_ baited traps [20]. The results provided evidence that the ovi-sticky traps were ineffective under local conditions, despite attempts to optimize their effectiveness. This contrasts with the results obtained in Singapore, where ovi-sticky traps have been used successfully for many years to monitor the two *Aedes* species [22,23]. This can be explained by differences in urban management and vector control strategies between the two islands. In Singapore, the prevention and elimination of mosquito breeding sites is strictly regulated. Both private individuals and professionals must ensure that there are no breeding sites that are potentially attractive for mosquitoes, and will have to pay a fine if a breeding site is discovered by competent bodies. In addition, traps are mostly placed inside buildings. However, in La Reunion, although the official vector control strategy promotes social mobilization to prevent and eliminate mosquito breeding sites, many domestic breeding sites remain widespread and difficult to access, resulting in competition with ovi-sticky traps and reducing their attractiveness. In fact, although *Ae. albopictus* populations are structured by habitat in La Reunion [33], their genetic diversity is high [34], favoring the support of a wide range of possible breeding sites and, consequently, a diluting effect when a particular type of ovitrap is used to catch adult mosquitoes. In contrast, a single strain of *Ae. aegypti* is found in La Reunion. It is genetically closer to *Ae. mascarensis* (a mosquito endemic to Mauritius) than any other strain of *Ae. aegypti* and quite different from Asian strains [35,36]. Its ecological niche, including favorable breeding sites, is much narrower than for *Ae. albopictus* [4,6,24]. Obviously, the settings of the ovi-sticky traps did not meet these requirements.

Adding BG-lure to CO_2_-baited traps did not improve their attractiveness for female mosquitoes, regardless of their species. This contrasts with other studies [37]. Such discrepancies could be explained by bioecological differences between mosquito populations or by environmental heterogeneity. To assess which bait will significantly improve the attractiveness of a given trap for a specific mosquito population, a field trial is, thus, required. Given the financial and logistic costs associated with the use of BG-Lure, and in the absence of robust statistical evidence, the best option for long-term monitoring implemented as part of the OpTIS project in the context of Saint-Joseph remains the use of BGS traps baited with CO_2_ alone.

We observed considerable variability in trap performance depending on weather and location within the study area. For example, before Cyclone Garance (27–28 February 2025), 169 adult mosquitoes were captured in the Langevin orchard (12 trapping sessions of 24 h), while fewer than 10 were collected in gardens during heavy rains (64 trapping sessions of 48 h).

Non-target species such as chironomid larvae and adult *Culex quinquefasciatus*, a potential vector of Rift Valley fever and West Nile viruses (both circulating in the south-western Indian Ocean), were also captured [38]. It is not surprising that *Ae. aegypti* and the chironomid larvae were positively correlated, since the *Ae. aegypti* strain from La Reunion is selvatic [4]. Although not central to this study, their presence provides valuable information for future environmental impact assessments related to the release of sterile mosquito males treated with pyriproxyfen as part of the OpTIS project.

The sudden drop in the *Ae. aegypti* density after Cyclone Garance suggests that extreme weather events may significantly affect their populations, especially considering their typical larval habitats in ravines, which consist mainly of rock hollows (data not presented in this study) [4,6]. This was also observed in Mauritius during a pilot SIT trial [13].

Larval and adult collections confirmed that *Ae. albopictus* was the dominant species in Saint-Joseph, as previously observed [20,24,33]. More surprisingly, *Ae. aegypti* larvae were also present in urban domestic environments. On some occasions, peak activity of adults was observed (Figure 4), outnumbering the *Ae. albopictus* density in this environment. These findings are of interest to public health and highlight a possible strengthening of virus transmission by *Ae. aegypti* during outbreaks of arboviruses.

Rainfall patterns could also play a role in the relative abundance of *Ae. aegypti* and *Ae. albopictus* (Figure 5). The mosquito productivity of natural breeding sites of *Ae. aegypti* (e.g., vacoas) is likely to depend more on rainfall than on human activity, in contrast to the mosquito productivity in artificial containers for which *Ae. albopictus* is better adapted than *Ae. aegypti* in La Reunion.

However, the distribution data for *Ae. aegypti* remain preliminary. Longitudinal studies covering different seasons and habitats (e.g., ravines vs. households) are needed to better understand the spatial and seasonal dynamics of both *Aedes* species. More generally, it seems important to determine the ecological niche and the actual distribution of *Ae. aegypti* in sympatry with *Ae. albopictus*. This knowledge would allow for better tailoring of control measures to local ecological contexts. Furthermore, future research comparing the vector capacity of *Ae. aegypti* and *Ae. albopictus* under the La Reunion conditions would be valuable, including vector competence studies already conducted or planned for these two species for dengue, chikungunya, and Zika viruses, which pose a risk of introduction and spread to the region [39].

We found *Ae. albopictus* in four of five breeding habitats where *Ae. aegypti* was recorded. This is promising with respect to the cross-efficacy of future control trials using boosted SIT with sterile male *Ae. albopictus* to carry pyriproxyfen. Recently, we proposed to use the heterospecific efficacy of the boosted SIT to prevent the invasion of new territories by invasive species [40].

This study has several limitations. First, the limited number of Latin square repetitions, which may restrict the generalization of the findings. A longer sampling period covering different seasons would provide more representative and robust data. Second, unmeasured factors, such as fine-scale meteorological fluctuations [41] or human activity patterns, could have influenced the efficacy of the trap and the presence of species. In addition, a chikungunya outbreak started in 2024 in La Reunion and virus transmission was still active when this study started [9]. Consequently, vector control operations occurred in the study area, consisting of deltamethrin fumigation targeting adult mosquitoes, destruction of breeding sites, and community participation [42]. Although special attention was paid to avoid the implementation of field work during vector control operations, these measures probably had an effect on the density of *Ae. albopictus* and *Ae. aegypti*. Finally, conclusions drawn from the monitoring of larval sites should be considered with caution. In fact, the small sample size (n=10, including six vacoas) makes the results unstable and the statistical power of the analysis low. Rather than focusing on the larval data, we tried to combine them with adult data to obtain a broader picture of the distribution and dynamics of mosquito populations.

## 5. Conclusions

This study highlights the importance of taking into account local specificity when selecting trapping methods. BGS traps baited with CO_2_ alone will be used for mosquito monitoring in the OpTIS project. The findings on the distribution of *Ae. aegypti* populations remain preliminary data. It would be necessary to study their distribution not only near ravines, but also within households, taking into account weather conditions during a longitudinal study that spans different seasons, and to compare it with the distribution of *Ae. albopictus* populations. A better understanding of the spatial distribution and seasonal dynamics of local target species can lead to more effective control strategies.

## Figures and Tables

**Figure 1 insects-16-00932-f001:**
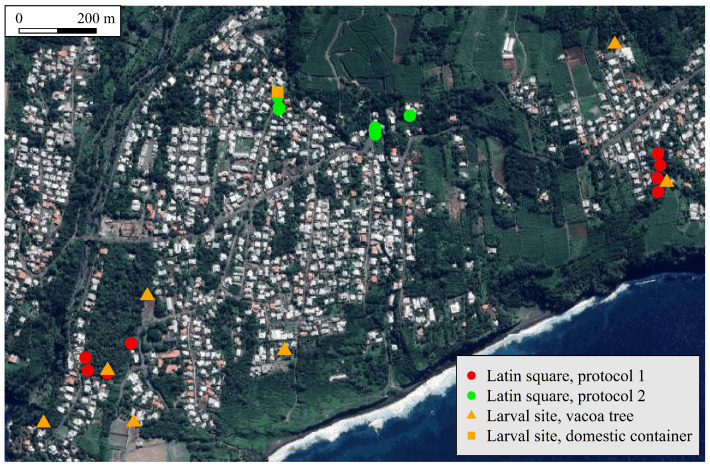
Trap location set for a field experiment using with a Latin square design and for a larval habitat stud, both conducted in Saint-Joseph (La Reunion) from February to May 2025. The background image was retrieved from the Google Maps Platform https://mapsplatform.google.com/ using functions available in the ggmap package for R https://github.com/features/packages/ggmap version 4.0.2 (accessed on 2 September 2025), together with a private API key.

**Figure 2 insects-16-00932-f002:**
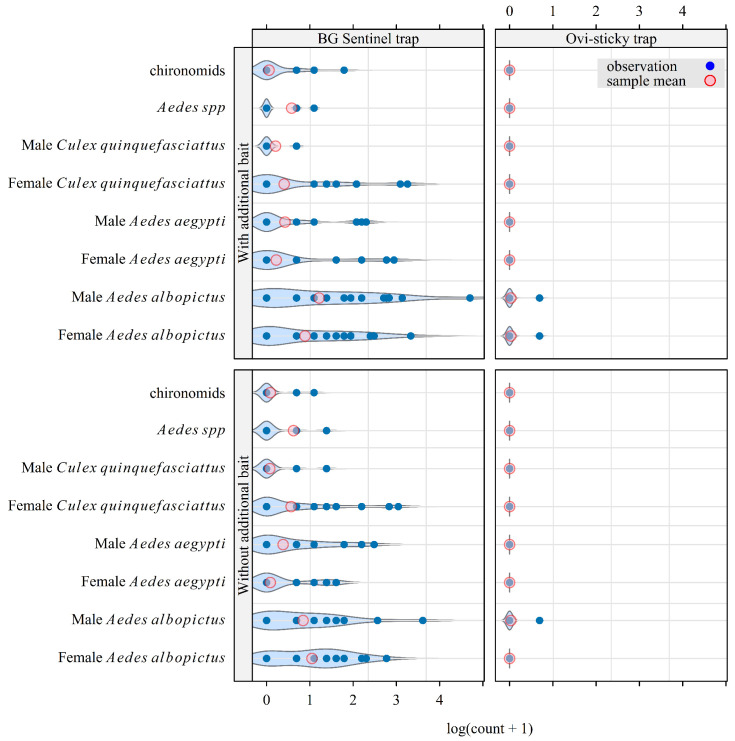
Cumulative catches for a selection of insects collected during the field study in Saint-Joseph (La Reunion) from February to May 2025 to compare the attractiveness of CO_2_-baited BG Sentinel traps and ovi-sticky traps, each under two different bait options. A Latin square design was used, with two protocols, each comprising two sets of four traps in two locations, and four and eight replicates/traps in protocols 1 and 2, i.e., 32 trapping sessions of 24 h, and 64 trapping sessions of 48 h, respectively. The violin plot in the background of each series of points is an estimate of the probability density of that series.

**Figure 3 insects-16-00932-f003:**
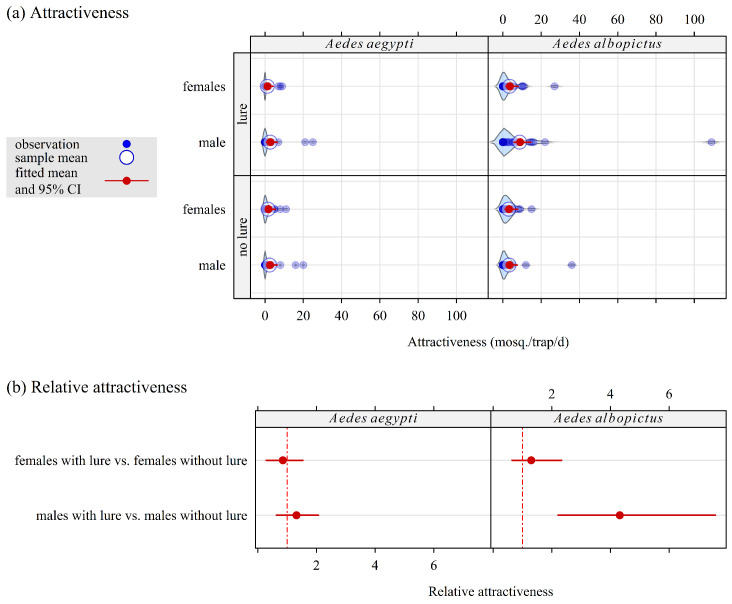
Attractiveness and relative attractiveness of CO_2_-baited traps fitted with a Bayesian mixed-effect Poisson model for *Ae. aegypti* (left panel) and *Ae. albopictus* (right panel) from a field study conducted in Saint-Joseph (La Reunion) from February to May 2025. A Latin square design (two protocols, each comprising two sets of four traps in two locations four and eight replicates/traps in protocols 1 and 2, i.e., 32 trapping session of 24 h, and 64 trapping sessions of 48 h, respectively) to evaluate the attractiveness obtained with the BG lure. The violin plot in the background of each series of points is an estimate of the probability density of this series.

**Figure 4 insects-16-00932-f004:**
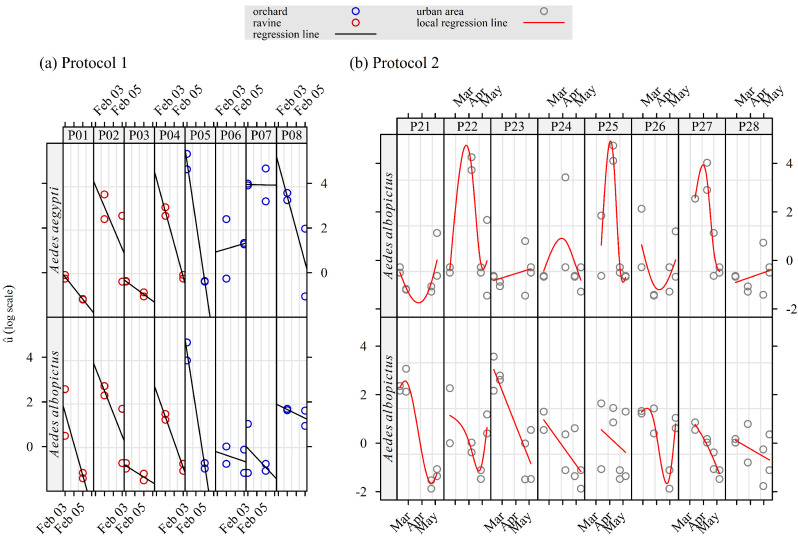
Observation-level random effect $\hat{u}$ estimated with a Bayesian mixed-effect Poisson model of trap attractiveness for *Aedes aegypti* and *Aedes albopictus* in a field study implemented in Saint-Joseph (La Reunion) from February to May 2025. Two distinct protocols were used, each with a Latin square design: (**a**) protocol 1: Two sets of four traps in two contrasted locations, four replicates, i.e., 32 trapping sessions of 24 h each; (**b**) protocol 2: Two sets of four traps in two similar locations, eight replicates/trap, i.e., 64 trapping sessions of 48 h each.

**Figure 5 insects-16-00932-f005:**
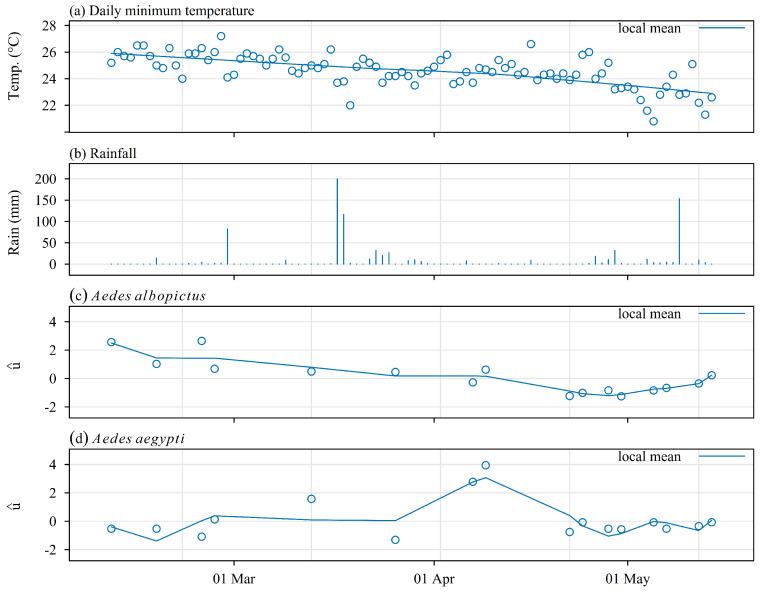
Meteorological conditions ((**a**) daily minimum temperature; (**b**) daily rainfall) and time trend in the random effect ((**c**) *Ae. albopictus*; (**d**) *Ae. aegypti*) estimated using a Bayesian mixed-effect Poisson model of adult mosquito attractiveness for CO_2_-baited traps. The random effect shown on this plot was estimated with data from protocol 2: Latin square design, two sets of four traps in two similar locations, 8 replicates from March to May 2025, i.e., 64 trapping sessions of 48 h each. Source of meteorological data: Meteo-France, Saint-Joseph station.

**Figure 6 insects-16-00932-f006:**
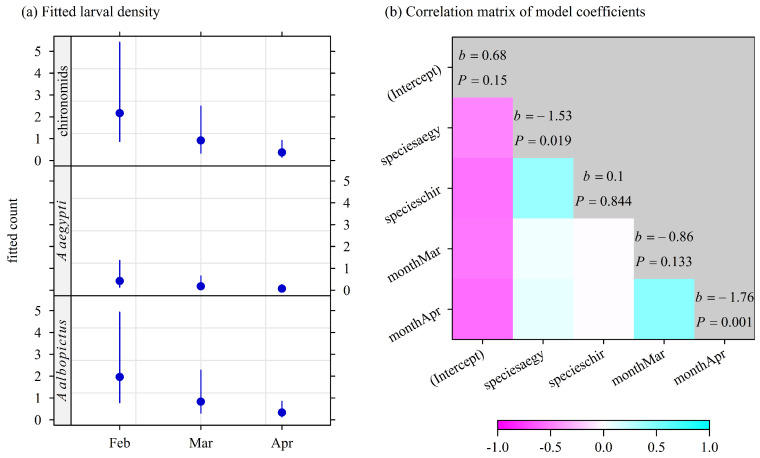
Larvae density (larvae count/leaf) in the water sampled in the stipes of vacoa trees in Saint-Joseph, February–April 2025. (**a**) Density fitted with a negative binomial model. (**b**) Correlation matrix of model coefficients: correlations were mapped according to the color key; on the matrix diagonal, *b* is the fitted coefficient and *P* is the *p*-value of the Wald test with $H_{0}$: *b* = 0. The intercept was the log of fitted density of *Ae. albopictus* larvae in February. Sample size: n=6 vacoas sampled on 4 occasions.

## Data Availability

The original data and R code presented in the study are openly available in Figshare at https://figshare.com/s/3b71db503502dfca6fae (accessed on 2 September 2025), DOI 10.6084/m9.figshare.29616800.

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
