# Peer review of "Distribution of Larval Habitats and Efficiency of Various Trap Settings to Monitor Sympatric Aedes albopictus and Aedes aegypti in La Reunion"

_insects, 2025, doi:10.3390/insects16090932_

Round 1
Reviewer 1 Report
Comments and Suggestions for Authors
Overall, a decent paper.
General comments: Have you considered using ovicups for surveillance? These are cheaper to deploy and can be useful for surveillance of both species of interest. It has the added benefit of storing the eggs until you are ready to hatch them.
Was there any systematic adult of larval mosquito control operations occurring during your study period? This could impact surveillance numbers.
Did you test any mosquitoes for pathogens?
Specifics:
Lines
48-49 citation
81 – does tap water include chlorine? If so was the water dechlorinated prior to putting it out?
98- ovi-sticky
117 – suggest you not freeze, but use ~70% ethanol
119 – why did you choose L3 stage for identification.
274 -sylvatic?
Author Response
General comments: Have you considered using ovicups for surveillance? These are cheaper to deploy and can be useful for surveillance of both species of interest. It has the added benefit of storing the eggs until you are ready to hatch them.
- Yes, we will use this system for monitoring egg density & hatching rate during the boosted SIT trial. However, ovi-sticky traps were used for the surveillance of adult mosquitoes. We added this sentence to the discussion L275: “Standard ovitraps will also be used to monitor egg density and hatching rate.”
Was there any systematic adult of larval mosquito control operations occurring during your study period? This could impact surveillance numbers.
- Yes indeed. We added this paragraph at the beginning of the discussion:
As indicated in the Introduction, a chikungunya outbreak started in 2024 in La Reunion and virus transmission was still active when this study started [9]. Consequently, vector control operations occurred in the study area, consisting of deltamethrin fumigation targeting adult mosquitoes, destruction of breeding sites, and community participation [26]. Although we were careful to avoid the implementation of field work during vector control operations, these measures probably had an effect on the density of Ae. albopictus and Ae. aegypti.
Did you test any mosquitoes for pathogens?
- No, we did not.
Specifics:
Lines
48-49 citation
81 – does tap water include chlorine? If so was the water dechlorinated prior to putting it out?
Yes, tap water is treated with chlorine in Saint-Joseph. It was systematically dechlorinated. To this end, trap water to be used with mosquitoes was first poured into a dedicated container which was left open during 48 hours at ambient temperature. It is now specified L131.
98- ovi-sticky
117 – suggest you not freeze, but use ~70% ethanol
- Thank you for the suggestion.
119 – why did you choose L3 stage for identification.
We chose L3/L4 larvae for identification because it is more uncertain in younger larvae.
274 -sylvatic?
- Selvatic.
Reviewer 2 Report
Comments and Suggestions for Authors
The manuscript addresses a timely and practical question relevant to vector control operations in La Reunion. It provides valuable data for optimizing entomological monitoring in preparation for a boosted Sterile Insect Technique (SIT) trial targeting Aedes albopictus. The study design is generally appropriate, and conclusions are mostly supported by the data. Minor revisions are required to clarify methodological details and contextualize findings.
Methodology:
The Latin square design is robust for comparing traps. However, the switch from 24-hour rotations (Protocol 1) to weekly rotations (Protocol 2) for ovi-sticky traps lacks justification. Potential bias due to unequal sampling effort should be discussed.
Sampling only 10 breeding sites (4 domestic, 6 vacoas) with 101 total larvae limits statistical power. The negative/positive correlations between species in vacoas (Fig. 6) are intriguing but warrant caution due to low sample size. Clarify how "density" was quantified (larvae per leaf? per mL?).
The Bayesian approach is appropriate. However, the model for larval co-occurrence (negative binomial regression) should explicitly state how overdispersion was addressed.
Discussion & Limitations:
The contrast with Singapore’s success is well-discussed, but differences in trap deployment (e.g., placement height, lure concentration) should be explored.
Address why results contradict Wilke et al. (2019), potentially citing local mosquito strain differences or environmental interference.
Acknowledge that limited sampling (10 sites, 4 sessions) precludes broad generalizations about habitat partitioning.
Minor Comments:
Introduction: Briefly define "boosted SIT" for non-specialist readers.
Methods: Specify the volume of water sampled per vacoa stipe.
Results: Report total trap-nights for each protocol.
References: Include recent SIT studies in Réunion.
Author Response
The manuscript addresses a timely and practical question relevant to vector control operations in La Reunion. It provides valuable data for optimizing entomological monitoring in preparation for a boosted Sterile Insect Technique (SIT) trial targeting Aedes albopictus. The study design is generally appropriate, and conclusions are mostly supported by the data. Minor revisions are required to clarify methodological details and contextualize findings.
To address the sub-optimal evaluation of our manuscript by this reviewer, we revised extensively this manuscript:
- We added information and references, both in the introduction and the discussion,
- We improved the results with a more comprehensive, compact, consistent and shorter set of plots (6 vs 8 in the initial manuscript), without loss of information. We also found, and corrected a mistake in the estimated relative attractiveness, without consequences on the conclusions (fig 3, formerly fig. 3 and 4).
- We shortened and re-organised the discussion, avoiding over-interpretation of the presented data.
Methodology:
The Latin square design is robust for comparing traps. However, the switch from 24-hour rotations (Protocol 1) to weekly rotations (Protocol 2) for ovi-sticky traps lacks justification. Potential bias due to unequal sampling effort should be discussed.
This sentence was added in the material and methods section:
In addition, the logarithm of the duration (in days) of each trapping session was used as an offset term in the mixed-effect Poisson model to account for differences in sampling effort between protocols 1 and 2.
Sampling only 10 breeding sites (4 domestic, 6 vacoas) with 101 total larvae limits statistical power. The negative/positive correlations between species in vacoas (Fig. 6) are intriguing but warrant caution due to low sample size. Clarify how "density" was quantified (larvae per leaf? per mL?).
- We added an introductory paragraph as a disclaimer:
Conclusions drawn from the monitoring of larval sites should be considered with caution. In fact, the small sample size (n=10, including 6 vacoas) makes the results unstable and the statistical power of the analysis low. Rather than focusing on the larval data, we tried to combine them with adult data to obtain a broader view of mosquito distribution and population dynamics
- Accordingly, we restructured the discussion and introduced two sub-sections, namely (1) To bait or not to bait and (2) Spatial distribution of Aedes aegypti and Aedes albopictus.
- The unit is larvae / leaf (15-50 ml / leaf). This has been made explicit in the section Material and methods, in plot legend (fig 4; ex fig. 5-6) and the corresponding text.
The Bayesian approach is appropriate. However, the model for larval co-occurrence (negative binomial regression) should explicitly state how overdispersion was addressed.
- We added a paragraph with references explaining the negative binomial distribution and its overdispersion parameter.
Discussion & Limitations:
The contrast with Singapore’s success is well-discussed, but differences in trap deployment (e.g., placement height, lure concentration) should be explored.
- In our opinion it is not possible to compare trap deployment in Singapore and La Réunion because the environments and local mosquito ecology are completely different. Instead, we explained the specificities of mosquito populations in La Réunion. Also, we added this sentence in the discussion: “Also, traps are mostly set inside buildings.”
Address why results contradict Wilke et al. (2019), potentially citing local mosquito strain differences or environmental interference.
- In our opinion, the same remark applies. We just have to assess which bait works or not. We added this sentence after Wilke citation:
In other words, it looks quite difficult to predict which bait is going to significantly improve the attractiveness of a given trap for a specific mosquito population: it has to be assessed with a field trial.
Acknowledge that limited sampling (10 sites, 4 sessions) precludes broad generalizations about habitat partitioning.
- Done: see above.
Minor Comments:
Introduction: Briefly define "boosted SIT" for non-specialist readers.
- Done
Methods: Specify the volume of water sampled per vacoa stipe.
- Done
Results: Report total trap-nights for each protocol.
- Done in the plot legends
References: Include recent SIT studies in Réunion.
- Done